# Light Alcohol Consumption Promotes Early Neurogenesis Following Ischemic Stroke in Adult C57BL/6J Mice

**DOI:** 10.3390/biomedicines11041074

**Published:** 2023-04-02

**Authors:** Jiyu Li, Chun Li, Pushpa Subedi, Xinli Tian, Xiaohong Lu, Sumitra Miriyala, Manikandan Panchatcharam, Hong Sun

**Affiliations:** 1Department of Cellular Biology & Anatomy, LSUHSC-Shreveport, Shreveport, LA 71103, USA; 2Department of Pharmacology, Toxicology & Neuroscience, LSUHSC-Shreveport, Shreveport, LA 71103, USA

**Keywords:** alcohol, brain, neurogenesis, subventricular zone, dentate gyrus, ischemia, reperfusion, rotarod test, open field test, mice

## Abstract

Ischemic stroke is one of the leading causes of death and disability worldwide. Neurogenesis plays a crucial role in postischemic functional recovery. Alcohol dose-dependently affects the prognosis of ischemic stroke. We investigated the impact of light alcohol consumption (LAC) on neurogenesis under physiological conditions and following ischemic stroke. C57BL/6J mice (three months old) were fed with 0.7 g/kg/day ethanol (designed as LAC) or volume-matched water (designed as control) daily for eight weeks. To evaluate neurogenesis, the numbers of 5-bromo-2-deoxyuridine (BrdU)^+^/doublecortin (DCX)^+^ and BrdU^+^/NeuN^+^ neurons were assessed in the subventricular zone (SVZ), dentate gyrus (DG), ischemic cortex, and ischemic striatum. The locomotor activity was determined by the accelerating rotarod and open field tests. LAC significantly increased BrdU^+^/DCX^+^ and BrdU^+^/NeuN^+^ cells in the SVZ under physiological conditions. Ischemic stroke dramatically increased BrdU^+^/DCX^+^ and BrdU^+^/NeuN^+^ cells in the DG, SVZ, ischemic cortex, and ischemic striatum. The increase in BrdU^+^/DCX^+^ cells was significantly greater in LAC mice compared to the control mice. In addition, LAC significantly increased BrdU^+^/NeuN^+^ cells by about three folds in the DG, SVZ, and ischemic cortex. Furthermore, LAC reduced ischemic brain damage and improved locomotor activity. Therefore, LAC may protect the brain against ischemic stroke by promoting neurogenesis.

## 1. Introduction

One of the leading global causes of death and permanent disability is stroke [1,2]. Ischemic strokes, which account for 87% of all strokes, result from an obstruction or narrowing in an artery that carries blood to the brain [3]. Two reperfusion therapies, pharmaceutical dissolution and mechanical blood clot removal, are the only ones currently approved for treating acute ischemic stroke [4,5]. Thus, transient focal cerebral ischemia has become one of the most common types of ischemic stroke. Unfortunately, although reperfusion/recanalization is critical for limiting ischemic brain damage, it may paradoxically worsen brain damage by inducing reperfusion injury [1,6]. Thus, most stroke survivors retain a variety of neurological deficits resulting from brain ischemia/reperfusion (I/R) damage. Therefore, it is imperative to develop novel therapeutic strategies to prevent and treat brain I/R damage [7,8,9].

After an ischemic stroke, endogenous regeneration occurs in the ischemic area [10,11]. Although the ability of intrinsic self-healing of the brain is limited, some neurological deficits, especially motor ones, show a spontaneous recovery during the chronic phase [12]. Thus, promoting endogenous regeneration may be an effective strategy to improve postischemic functional recovery. Evidence suggests that neurogenesis persists in the adult mammalian brain under physiological and pathological conditions [13,14,15]. The subventricular zone (SVZ) of the lateral ventricles and the subgranular zone (SGZ) of the dentate gyrus (DG) are the neurogenic regions in the adult brain [16,17]. Neuroblasts differentiated from the SVZ and SGZ migrate to the olfactory bulb, local parenchyma, and granule cell layer of the DG under physiological conditions [18]. After ischemic stroke, the proliferation and differentiation of neuronal progenitors dramatically increase. The neuroblasts formed before and after the stroke migrate to the lesion area and differentiate into functional neurons [19,20,21]. Postischemic neurogenesis starts as early as two days, peaks at two weeks, and continues for more than six weeks after the onset of ischemic stroke [15,22,23]. Although the precise mechanism driving postischemic neurogenesis is still unclear, many factors, including growth factors and inflammatory modulators, have been reported to be involved in the process [15,24,25].

Alcohol is one of the most frequently used chemicals. Its actions frequently target the brain [26]. Various epidemiological studies suggest that chronic alcohol consumption has dose-dependent effects on the incidence and prognosis of ischemic stroke. In contrast to heavy alcohol consumption (HAC), which increases the incidence and worsens the outcome of ischemic stroke, light-to-moderate alcohol consumption (LAC) lowers the incidence and improves the prognosis of ischemic stroke [27,28,29,30,31,32,33,34,35]. Recently, we discovered that LAC protected against brain I/R damage by promoting cerebral angiogenesis and reducing postischemic apoptosis, inflammation, and blood-brain barrier (BBB) disruption in rodents [36,37,38]. However, research studies that we are aware of have yet to investigate the impact of LAC on neurogenesis following ischemic stroke. In addition, the effect of chronic alcohol consumption on baseline neurogenesis has yet to be studied to a large extent. Previous studies focused on the impact of alcohol consumption on neurogenesis in the DG of the hippocampal formation [39,40,41]. However, the results lack consistency. Thus, our goal of the present study is to ascertain whether LAC impacts neurogenesis under physiological conditions and following transient focal cerebral ischemia.

BrdU^+^/DCX^+^ and BrdU^+^/NeuN^+^ cells, which represent newborn neurons, are commonly used to evaluate neurogenesis [42,43,44,45]. In the present study, we compared the number of BrdU^+^/DCX^+^ and BrdU^+^/NeuN^+^ neurons in the DG, SVZ, ischemic cortex, and ischemic striatum between LAC mice and control mice under physiological conditions and two weeks after transient focal cerebral ischemia. A previous study found that LAC reduced infarct size and improved motor function at an early reperfusion stage [37]. However, the impact of LAC on ischemic damage and locomotor activity has not been assessed at the late stage of reperfusion. Nissl staining is a valuable tool to determine postischemic brain atrophy [46,47]. On the other hand, the rotarod and open field tests are extensively used for analyzing locomotor activity [48,49,50]. Therefore, we further measured the effects of LAC on ischemic damage and locomotor activity at two weeks of reperfusion using the Nissl stain, rotarod test, and open field test.

## 2. Methods

### 2.1. Animal Models

The Louisiana State University Health Science Center (LSUHSC)-Shreveport Institutional Animal Care and Use Committee (IACUC) gave its approval to all the procedures and protocols in the present study, which were carried out following the ARRIVE (Animal Research: Reporting in Vivo Experiments) and National Institutes of Health (NIH) Guide for the Care and Use of Laboratory Animals. The protocol (P-21-038) was approved on 12 May 2021. Forty C57BL/6J mice (male, three months old, 25–30 g) were fed with 0.7 g/kg/day ethanol (designed as LAC, *n* = 20) or volume-matched water (designed as control, *n* = 20) through gavage once a day for eight weeks [36]. These mice were housed in Animal Resources at LSUHSC-Shreveport. The mice were housed in a room maintained at 20–22 °C and light cycle from 5:00 a.m. to 5:00 p.m. Body weight, blood pressure, heart rate, and fasting blood glucose level were measured at the end of the eight-week feeding period as previously described [36]. To measure fasting blood glucose level, mice fasted for 12 h during the daytime. In each group, five mice were used to assess neurogenesis under physiological conditions, five mice were used to determine the neurogenesis following transient focal cerebral ischemia, and the remaining ten mice were used to evaluate the locomotor activity under physiological conditions and following transient focal cerebral ischemia. To measure neurogenesis under physiological conditions, 50 mg/kg bromodeoxyuridine (BrdU) (B5002, Sigma-Aldrich, St. Louis, MI, USA) was intraperitoneally administered once a day for ten days from the fifth week. The mice were euthanized at the end of the eight-week feeding period.

### 2.2. Transient Focal Cerebral Ischemia

To determine postischemic neurogenesis and locomotor activity, transient focal cerebral ischemia was induced by unilateral middle cerebral artery occlusion (MCAO) for 60 min at the end of the eight-week feeding period as previously described [38]. Ethanol was not given on the day of or following the ischemic stroke to prevent the acute effect of alcohol and simulate in-hospital alcohol discontinuation. Isoflurane in a gas mixture that contained 30% oxygen and 70% nitrogen was used to induce (5%) and maintain (1.5%) the mouse’s anesthesia. During the procedure, a temperature-controlled heating pad (Harvard Apparatus, March, Germany) was used to keep the body temperature stable, and a Laser-Doppler flow probe (PERIMED, PF 5010 LDPM Unit, Järfälla, Sweden) was used to monitor the regional cerebral blood flow (rCBF) of the right MCA territory. To occlude the right MCA, the right external carotid artery (ECA) and common carotid artery (CCA) were exposed and ligated. Then, a silicon rubber-coated monofilament was inserted into the right internal carotid artery (ICA) from the base of the right ECA cranially to the bifurcation, where the ICA splits into the MCA and the anterior cerebral artery (ACA). A sharp decline in rCBF in the MCA territory indicated the initiation of MCAO. Following 60 min of occlusion, the monofilament was withdrawn, and the CCA was reopened to allow for reperfusion. To measure postischemic neurogenesis, 75 mg/kg BrdU was intraperitoneally administered twice daily for five consecutive days from 48 h of reperfusion. Mice were euthanized at 14 days of reperfusion.

### 2.3. Immunohistochemistry Staining

To determine the impact of chronic alcohol consumption on neurogenesis, dual immunohistochemistry staining was performed. The anesthetized mice were transcardially perfused with phosphate-buffered saline (PBS) and 4% paraformaldehyde. The brains were removed, fixed for an overnight period in 4% paraformaldehyde, dehydrated for 72 h in a graded series of sugar solutions, embedded for 5 min in O.C.T. compound (23-730-571, Fisher Scientific, Waltham, MA, USA), and then immediately frozen in liquid nitrogen. Coronal sections of 14 μm thickness were cut from the frozen brains and placed on frost-free slides. Three sections (0.25 mm rostral and 0.47 mm and 2.15 mm caudal to bregma) from each mouse were incubated in 2 M HCl for 1 h and neutralized in 0.1 M sodium borate buffer (pH 8.5) for 10 min. After being washed with PBS, the sections were blocked with a mixture of 1% bovine serum albumin (BSA), 0.3% trypsin, and 5% goat serum for 1 h and incubated with 1:100 mouse anti-BrdU antibody (347580; BD Bioscience, La Jolla, CA, USA) at 4 °C overnight. Subsequently, the sections were washed with PBS, incubated with 1:200 biotinylated goat antimouse lgG antibody (BA-9200; Vector Labs, Newark, CA, USA) for 1 h, then 1:200 streptavidin Alexa Fluor^TM^ 488 conjugate (s-32354; Thermo Fisher, Waltham, MA, USA) for 30 min. Following three washes, the sections were blocked with a mixture of 1% BSA, 0.3% trypsin, and 5% donkey serum for 1 h and then incubated with 1:100 rabbit antidoublecortin (DCX) (4604S; Cell Signaling, Danvers, MA, USA) or 1:100 rabbit anti-NeuN (MABN140; MilliporeSigma, Burlington, MA, USA) for 3 h. The sections were washed with PBS and incubated with 1:200 Alexa Fluor 546 conjugated donkey antirabbit IgG antibody (A10040; Thermo Fisher) for one hour. After three PBS washes, the sections were coated with a DAPI mounting medium (H-1800, VectorShield, Newark, CA, USA) and then observed under a fluorescence microscope (Nikon Eclipse Ts2). Five pictures from each region of interest in the SVZ, DG, ischemic cortex, and ischemic striatum were captured for quantitative analysis. BrdU- and DCX/NeuN-positive cells were counted, and their fold changes from the control were expressed.

### 2.4. Cresyl Violet Staining

The section at 0.47 mm caudal to bregma, which had the largest infarct at 24 h of reperfusion in the MCAO mouse model, was selected for Cresyl Violet staining to evaluate ischemic damage as described previously [37]. The sections were cleaned in xylene, dehydrated in ethanol, incubated in 0.01% Cresyl Violet acetate (C5042, Sigma-Aldrich) solution at 60 °C for 14 min, and mounted using xylene-a-based mounting media (8312-4, VWR, Radnor, PA, USA). The section was pictured under 1.0× magnification (Olympus) and analyzed using ImageJ. Instead of a complete lack of staining, which is defined as the infarct, a reduction in the volume of the ipsilateral hemisphere was observed. Thus, ischemic damage was represented by the ratio of the ipsilateral hemispheric volume to the contralateral hemispheric volume.

### 2.5. Locomotor Activity

The accelerating rotarod (LSI Letica Scientific Instruments, Barcelona, Spain) test was conducted as described previously to assess the induced motor activity [51]. Briefly, mice were trained to stay on the rotarod at a constant speed of 4 rpm for three consecutive days during the eighth week of gavage feeding and then underwent three test trials per day with the rotarod set at an acceleration rate of 4–40 rpm/10 min at the end of the eighth week of the feeding period and three days, seven days, and fourteen days of reperfusion following 60-min MCAO. The latency to fall from the accelerating rotarod was recorded for each trial. The average of the three trials on each day was used for statistical analysis.

To assess spontaneous motor activity, the open field test was performed. After habituation to the testing room, mice were placed into a square open field chamber (40 cm L × 40 cm W × 30 cm H) (AccuScan Instruments, Erie, PA, USA). Mice were allowed to explore freely for 30 min. The mice’s movements within the chambers were recorded. The total distance traveled and the number of moves were analyzed using the Top Scan Lite-Top View Behavior Analyzing System (Noldus Information Technology, Wageningen, Gelderland, The Netherlands).

### 2.6. Statistical Analysis

The statistical analysis was done with Prism 9. The comparison of two independent groups was performed using an unpaired *t*-test. Means and the standard deviation (SD) are used to present the quantitative data. The differences are considered statistically significant when the *p*-value is less than 0.05.

## 3. Results

### 3.1. Control Conditions

Similar to the previously reported information, gavage feeding with 0.7 g/kg/day ethanol once a day for eight weeks did not significantly change body weight, mean arterial blood pressure (MABP), heart rate, or fasting glucose level (Table 1) [36,37].

### 3.2. Effect of LAC on Neurogenesis under Physiological Conditions

BrdU^+^/DCX^+^ and BrdU^+^/NeuN^+^ cells were observed in the SVZ and DG under physiological conditions (Figure 1A,C and Figure 2A,C) (Appendix A for higher magnification). Interestingly, an eight-week daily intake of 0.7 g/kg/day ethanol significantly increased the number of BrdU^+^/DCX^+^ and BrdU^+^/NeuN^+^ cells in the SVZ by two folds and six folds, respectively (Figure 1B,D). In addition, 0.7 g/kg/day of ethanol tended to increase the number of BrdU^+^/DCX^+^ and BrdU^+^/NeuN^+^ cells in the DG (Figure 2). However, the increase did not reach statistical significance compared to the control (Figure 2B,D).

### 3.3. Effect of LAC on Brain I/R Damage

Cresyl violet staining was performed to evaluate transient focal cerebral ischemia-induced brain damage. As shown in Figure 3A, a complete lack of staining, defined as the infarct lesion at 24 h of reperfusion [37,52], was not observed at 14 days of reperfusion following 60-min MCAO. Instead, a reduction in the volume of the ipsilateral (right) hemisphere was found. The ratio of the ipsilateral hemisphere to the contralateral hemisphere was significantly greater in 0.7 g/kg/day ethanol-fed mice compared to the control mice (Figure 3B).

### 3.4. Effect of LAC on Neurogenesis Following Transient Focal Cerebral Ischemia

Sixty-minute MCAO significantly increased BrdU^+^/DCX^+^ and BrdU^+^/NeuN^+^ cells in the SVZ and DG at 14 days of reperfusion (Figure 4A,C and Figure 5A,C) (Appendix A for higher magnification). In addition, BrdU^+^/DCX^+^ and BrdU^+^/NeuN^+^ cells can be seen in the ischemic cortex and ischemic striatum (Figure 6A,C and Figure 7A,C). As shown in Figure 4B, Figure 5B, Figure 6B and Figure 7B, the number of BrdU^+^/DCX^+^ cells in all observed areas was significantly greater in 0.7 g/kg/day ethanol-fed compared to the control mice. In addition, 0.7 g/kg/day of ethanol significantly augmented the postischemic increase in BrdU^+^/NeuN^+^ cells by about three folds in the SVZ and DG (Figure 4D and Figure 5D). Furthermore, the number of BrdU^+^/NeuN^+^ cells in the ischemic cortex was significantly greater (by more than three folds) in 0.7 g/kg/day ethanol-fed mice compared to the control mice (Figure 6D). On the other hand, 0.7 g/kg/day of ethanol tended to increase the number of BrdU^+^/NeuN^+^ cells in the ischemic striatum. However, the increase did not reach statistical significance compared to the control (Figure 7D).

### 3.5. Effect of LAC on the Locomotor Activity

The accelerating rotarod and open field tests were conducted to assess the impact of LAC on locomotor activity. Interestingly, eight-week ingestion of 0.7 g/kg/day ethanol significantly strengthened the induced and spontaneous motor activities under physiological conditions (Figure 8). In order to allow the animals to recover from the surgical procedure of the MCAO for two days, both tests were not conducted for two days after ischemia. As shown in Figure 8A, the induced motor activity was significantly enhanced at 3 days, 7 days, and 14 days of reperfusion in 0.7 g/kg/day ethanol-fed mice compared to the control mice. In addition, 0.7 g/kg/day of ethanol significantly increased the total distance and number of movements in the open field test at seven days of reperfusion (Figure 8B,C).

In summary, LAC reduced postischemic brain atrophy at two weeks of reperfusion. Furthermore, LAC significantly increased newborn neurons in the SVZ, DG, and ischemic cortex and enhanced motor activity under physiological conditions and following ischemic stroke.

## 4. Discussion

The impact of LAC on neurogenesis under physiological circumstances and following transient focal cerebral ischemia was determined in the present study. There are several new findings. First, LAC increased neurogenesis in the SVZ but not DG under physiological conditions. Second, LAC alleviated transient focal cerebral ischemia-induced atrophy in the ipsilateral hemisphere at two weeks of reperfusion. Third, LAC augmented neurogenesis in the SVZ, DG, and ischemic cortex two weeks following transient focal cerebral ischemia. Fourth, LAC strengthened the locomotor activity under physiological conditions and following transient focal cerebral ischemia. We speculate that LAC may protect the brain against ischemic stroke by promoting neurogenesis. Thus, a future study appears necessary to elucidate the mechanism underlying LAC-induced neurogenesis, which may lead to new approaches for preventing and treating ischemic stroke and neurodegenerative diseases.

In our recent studies, eight-week daily consumption of 0.7 g/kg alcohol significantly diminished early ischemic brain damage in the mouse model of transient focal cerebral ischemia [36,37]. The peak blood alcohol concentration was about 9 mM [36], which can be seen following the consumption of 1.5 standard drinks (each containing 14 g of pure ethanol) in a male with average body weight [53]. Therefore, the dose of 0.7 g/kg/day ethanol in mice represents LAC in humans. BrdU, a thymidine analog readily incorporated into DNA during the S-phase of the cell cycle, is a reagent extensively used to label and quantify proliferating cells [54]. DCX is a nervous system-specific microtubule-associated protein (MAP) expressed in migrating neurons, and NeuN is a neuronal nuclear antigen expressed in mature neurons. Neural stem/progenitor cells (NSPCs) are present in the SVZ of the lateral ventricle and SGZ of the DG. After the ischemic process, NSPCs in these areas proliferate and migrate toward the lesion to participate in brain repair. Thus, the numbers of DCX^+^/BrdU^+^ and NeuN^+^/BrdU^+^ cells in the SVZ, DG, and ischemic areas of 0.7 g/kg/day ethanol-fed mice under physiological conditions and following ischemic stroke were counted to evaluate the impact of LAC on neurogenesis. In addition to mature neurons, NeuN is also expressed lightly in the maturing neurons [55,56]. Therefore, in the present study, both heavily and lightly NeuN-staining BrdU^+^ cells were counted as newborn neurons to indicate neurogenesis.

The brain is one of the primary target organs of alcohol’s effects. Several studies have examined the effects of acute and chronic alcohol exposure on hippocampal and subventricular neurogenesis under physiological conditions. However, the results appeared inconsistent. For example, an early study reported that acute alcohol intoxication dose-dependently inhibited NSPC proliferation in the DG and SVZ of male adolescent Sprague–Dawley rats [57]. Similarly, Taffe et al. found that long-term heavy alcohol consumption dramatically and persistently decreased hippocampal cell proliferation and neurogenesis in adolescent nonhuman primates [39]. Moreover, Liu and Crews found that adolescent intermittent ethanol exposure persistently decreased adult subventricular and hippocampal neurogenesis in Wistar rats [58]. Anderson et al. reported that chronic moderate alcohol consumption significantly decreased NSPC proliferation in the DG of either male or female adult Sprague-Dawley rats [40]. In contrast, Aberg et al. found that chronic moderate alcohol consumption improved hippocampal cell proliferation and neurogenesis in male adult C57BL/6 mice [41]. Xu et al. reported that two-month voluntary alcohol drinking stimulated neurogenesis in the DG and SVZ of male cHAP mice [59]. Recently, four-day binge drinking was shown to increase hippocampal neurogenesis in adult female Sprague-Dawley rats [60]. In the present study, although LAC tended to increase both DCX^+^/BrdU^+^ and NeuN^+^/BrdU^+^ cells in the DG, the increase did not reach statistically significant levels. However, LAC increased DCX^+^/BrdU^+^ cells by about two folds and NeuN^+^/BrdU^+^ cells by about six folds in the SVZ. The reason for the discrepancies between these studies regarding alcohol on hippocampal neurogenesis is unclear. It could be connected to the age, the length of alcohol exposure, the methods used to give alcohol, the timing of analysis, or the species differences in the effects of alcohol.

The present study is the first to examine the effects of LAC on postischemic neurogenesis. SVZ is the main source of neuroblasts generated following ischemic stroke [61,62,63]. It has been suggested that NSPCs in the SVZ proliferate, differentiate, and migrate to the infarct area, contributing to self-repair and repopulation of the injured area following ischemic stroke. Increasing evidence indicates that post-ischemic neurogenesis in the SVZ is involved in functional improvement [21,64]. On the other hand, the proliferation and differentiation of NSPCs in the DG are also remarkably stimulated following ischemic stroke. However, the production of new neurons in the DG influenced functional recovery negatively [15]. In the present study, LAC significantly increased both DCX^+^/BrdU^+^ and NeuN^+^/BrdU^+^ cells in the SVZ and DG. In addition, LAC significantly increased both DCX^+^/BrdU^+^ and NeuN^+^/BrdU^+^ cells in the ischemic cortex and DCX^+^/BrdU^+^ cells in the ischemic striatum. The ischemic injury’s severity is a major factor affecting postischemic neurogenesis. However, our previous studies have shown that LAC reduces cerebral I/R damage at 24 h of reperfusion in this mouse model of transient focal cerebral ischemia [36,37,65]. In the present study, LAC significantly reduced postischemic atrophy of the cerebral hemisphere at two weeks of reperfusion. Thus, it is conceivable that LAC preconditioning may favor postischemic endogenous repair, but not due to the compensatory response.

In addition to the severity of the brain injury, inflammation, growth factors, and angiogenesis also have been demonstrated to affect neurogenesis [15,24,25,66]. However, mechanisms that account for the effects of alcohol on neurogenesis under physiological conditions remain unclear. Our recent studies found LAC did not significantly alter that baseline DNA fragmentation. In addition, LAC did not lead to neuronal apoptosis under physiological conditions in adult mice [37,65]. Thus, it does not appear that the proneurogenic effect of LAC under physiological conditions is a response to a brain injury. In contrast, LAC may stimulate neurogenesis through its anti-inflammatory effect. LAC significantly changed the inflammatory profile in the brain under physiological conditions and following ischemic stroke [36]. LAC tends to increase anti-inflammatory cytokines/chemokines. It reduced IL-1beta and increased IL-1ra following ischemic stroke. Moreover, early postischemic microglia activation, which is a proinflammatory phenotype of microglia, was significantly suppressed by LAC. A recent study found that the blockade of the IL-1 receptor promoted the proliferation of NSPCs in the SVZ, enhanced neuroblast migration, and increased the number of newly born neurons in the ischemic cortex [67]. On the other hand, the effect of microglia activation on neurogenesis seems dual. An early study reported that microglia activation impaired hippocampal neurogenesis [68]. Recently, the anti-inflammatory phenotype of microglia was found to enhance the proliferation and differentiation of neuronal progenitors in the SVZ after ischemic stroke [69]. Moreover, it is also possible that LAC stimulates neurogenesis by upregulating the vascular endothelial growth factor (VEGF) and VEGF receptor 2 (VEGFR2). We recently found that LAC upregulates VEGF and VEGFR2 and promotes cerebral angiogenesis [38]. It has been suggested that VEGF/VEGFR2 signaling can directly lead to neurogenesis by stimulating cell proliferation [70]. In addition, the increased angiogenesis may facilitate neuroblasts to reach the damaged area by migrating along the newly formed blood vessels [66,71]. In our recent study, LAC significantly increased postischemic angiogenesis in the ischemic cortex [38]. Both DCX^+^/BrdU^+^ and NeuN^+^/BrdU^+^ cells in the ischemic cortex were consistently significantly greater in LAC mice. Thus, the precise mechanism by which LAC induces neurogenesis must be elucidated in the future.

In the present study, we further evaluated the impact of LAC on locomotor activity. The accelerating rotarod has been reliably used to test motor coordination and motor learning. The open field test is generally used to assess spontaneous locomotion [51]. Interestingly, LAC strengthened the locomotor activity in both tests under physiological conditions. However, it is unknown whether the enhanced locomotor activity resulted from the increased neurogenesis in LAC mice. In addition, although LAC significantly increased NeuN^+^/BrdU^+^ cells in the DG, SVZ, and ischemic cortex at two weeks of reperfusion, the total distance and number of movements of the open field test only reached statistically significant at one week, but not two weeks of reperfusion. Thus, future studies are essential to investigate the impacts of LAC-induced neurogenesis under physiological conditions and whether postischemic (in-hospital) alcohol cessation compromises the neuroprotective effect of LAC following ischemic stroke.

The present study provided additional evidence that chronic alcohol consumption significantly alters the pathophysiology of ischemic stroke. Although LAC appears beneficial to the prognosis of ischemic stroke in many aspects, alcohol, especially heavy alcohol consumption, is associated with cancer and other diseases [72,73,74]. Therefore, drinking alcohol is not encouraged.

## 5. Conclusions

The present study determined the impact of LAC on neurogenesis. LAC promoted neurogenesis under physiological conditions following transient focal cerebral ischemia. Thus, the present study merits further investigation. A better understanding of how alcohol impacts neurogenesis will not only improve the clinical treatment of ischemic stroke in alcohol users, but will also result in new approaches for preventing and treating ischemic stroke and neurodegenerative diseases in nondrinkers.

## Figures and Tables

**Figure 1 biomedicines-11-01074-f001:**
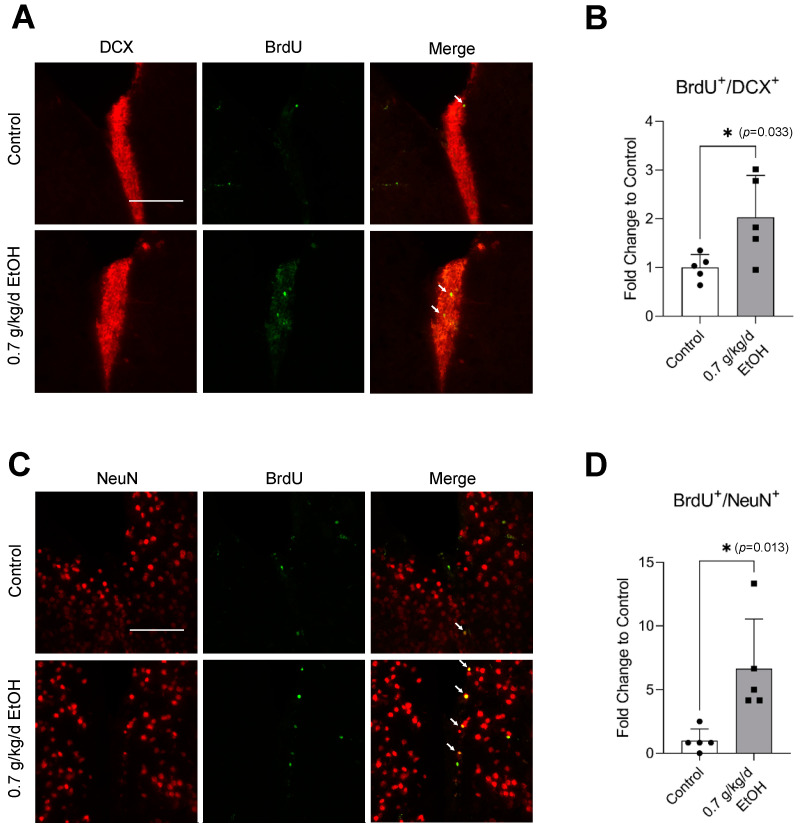
Effect of LAC on baseline neurogenesis in the SVZ. (**A**) Representative double staining of BrdU and DCX. Scale bar = 100 μm. (**B**) Values are means ± SD (*n* = 5). (**C**) Representative double staining of BrdU and NeuN. (**D**) Values are means ± SD (*n* = 5); * *p* < 0.05 vs. Control. Analyzed using an unpaired *t*-test.

**Figure 2 biomedicines-11-01074-f002:**
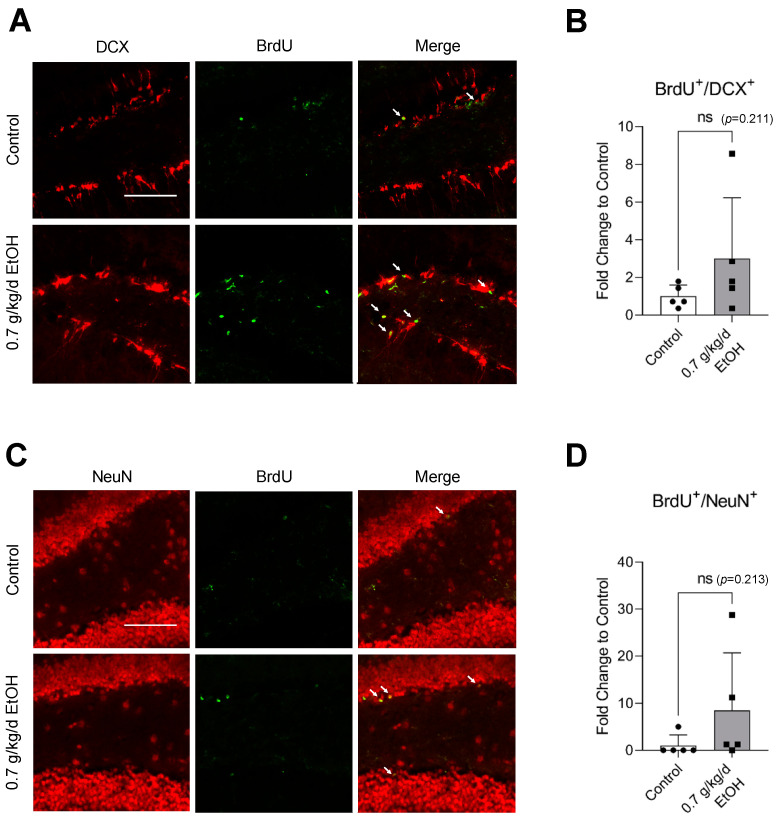
Effect of LAC on baseline neurogenesis in the DG. (**A**) Representative double staining of BrdU and DCX. Scale bar = 100 μm. (**B**) Values are means ± SD (*n* = 5). (**C**) Representative double staining of BrdU and NeuN. (**D**) Values are means ± SD (*n* = 5). Analyzed using an unpaired *t*-test.

**Figure 3 biomedicines-11-01074-f003:**
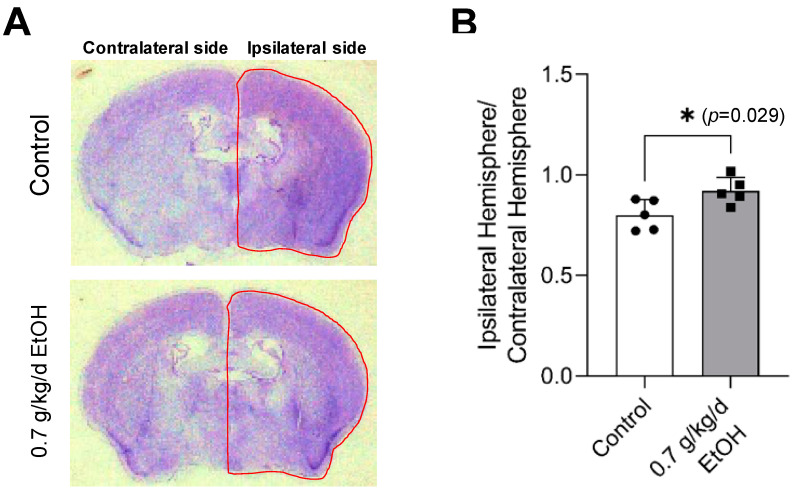
Effect of LAC on cerebral I/R damage at two weeks of reperfusion. (**A**) Representative brain sections stained with Cresyl violet. (**B**) The ratio of ipsilateral hemisphere/contralateral hemisphere. Values are means ± SD for five mice in each group; * *p* < 0.05 vs. Control. Analyzed using an unpaired *t*-test.

**Figure 4 biomedicines-11-01074-f004:**
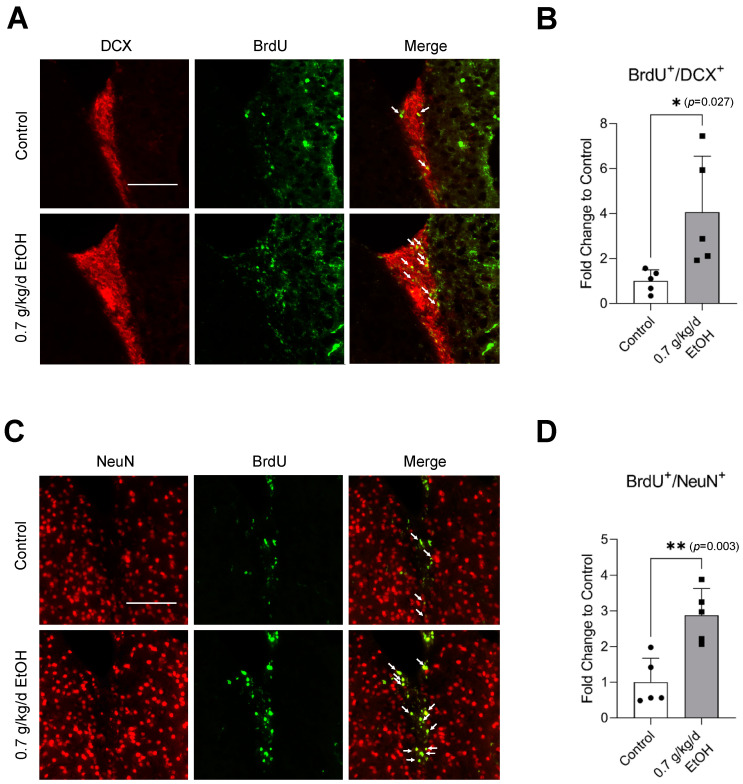
Effect of LAC on postischemic neurogenesis in the SVZ. (**A**) Representative double staining of BrdU and DCX. Scale bar = 100 μm. (**B**) Values are means ± SD (*n* = 5). (**C**) Representative double staining of BrdU and NeuN. (**D**) Values are means ± SD (*n* = 5); * *p* < 0.05; ** *p* < 0.01 vs. Control. Analyzed using an unpaired *t*-test.

**Figure 5 biomedicines-11-01074-f005:**
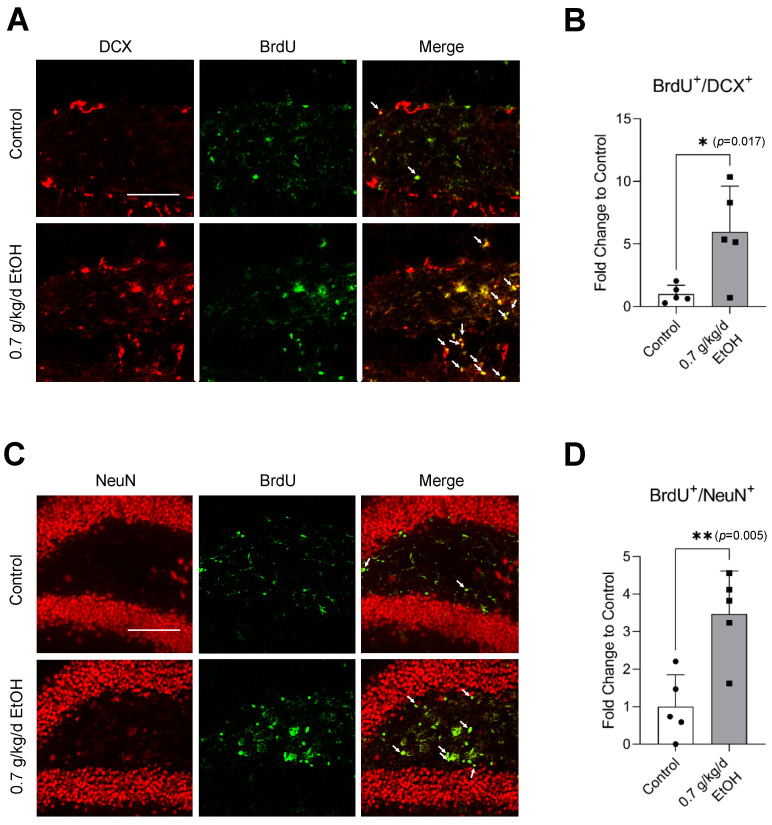
Effect of LAC on postischemic neurogenesis in the DG. (**A**) Representative double staining of BrdU and DCX. Scale bar = 100 μm. (**B**) Values are means ± SD (*n* = 5). (**C**) Representative double staining of BrdU and NeuN. (**D**) Values are means ± SD (*n* = 5); * *p* < 0.05; ** *p* < 0.01 vs. Control. Analyzed using an unpaired *t*-test.

**Figure 6 biomedicines-11-01074-f006:**
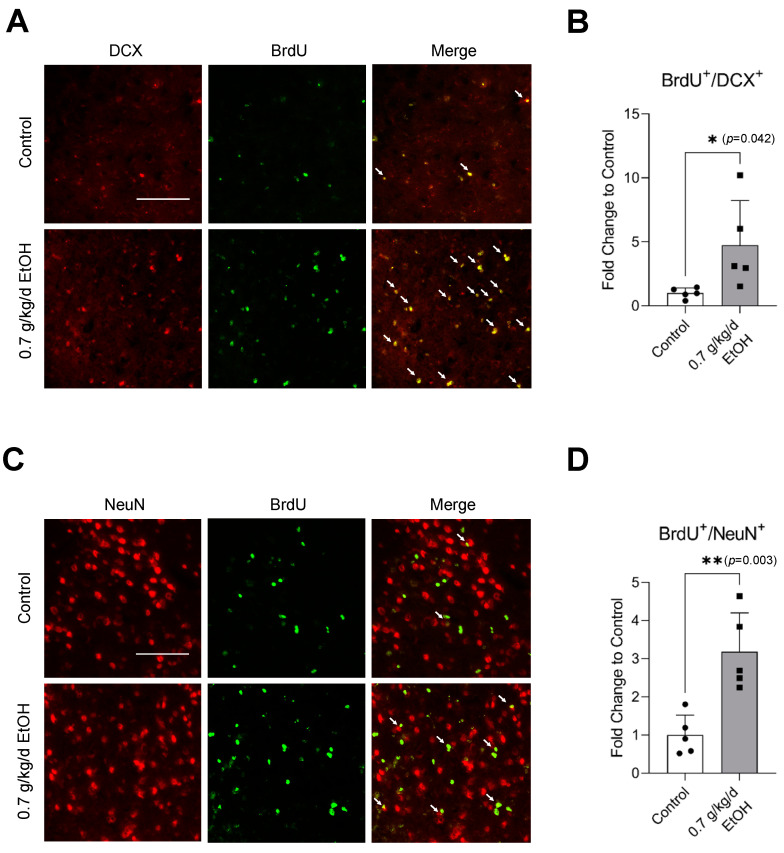
Effect of LAC on postischemic neurogenesis in the ischemic cortex. (**A**) Representative double staining of BrdU and DCX. Scale bar = 100 μm. (**B**) Values are means ± SD (*n* = 5). (**C**) Representative double staining of BrdU and NeuN. (**D**) Values are means ± SD (*n* = 5); * *p* < 0.05; ** *p* < 0.01 vs. Control. Analyzed using an unpaired *t*-test.

**Figure 7 biomedicines-11-01074-f007:**
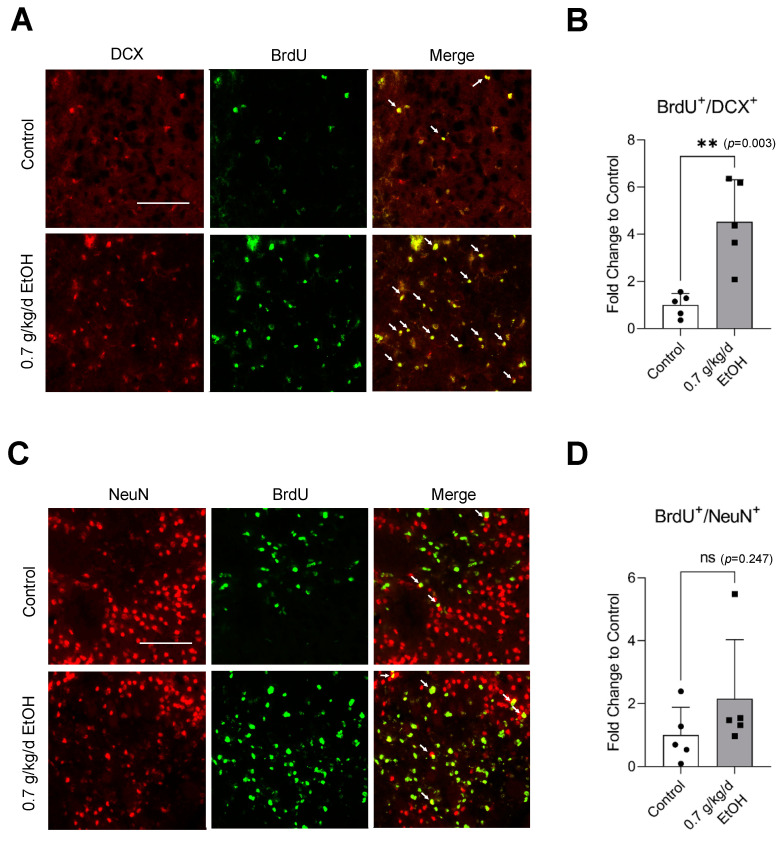
Effect of LAC on postischemic neurogenesis in the ischemic striatum. (**A**) Representative double staining of BrdU and DCX. Scale bar = 100 μm. (**B**) Values are means ± SD (*n* = 5). (**C**) Representative double staining of BrdU and NeuN. (**D**) Values are means ± SD (*n* = 5); ** *p* < 0.01 vs. Control. Analyzed using an unpaired *t*-test.

**Figure 8 biomedicines-11-01074-f008:**
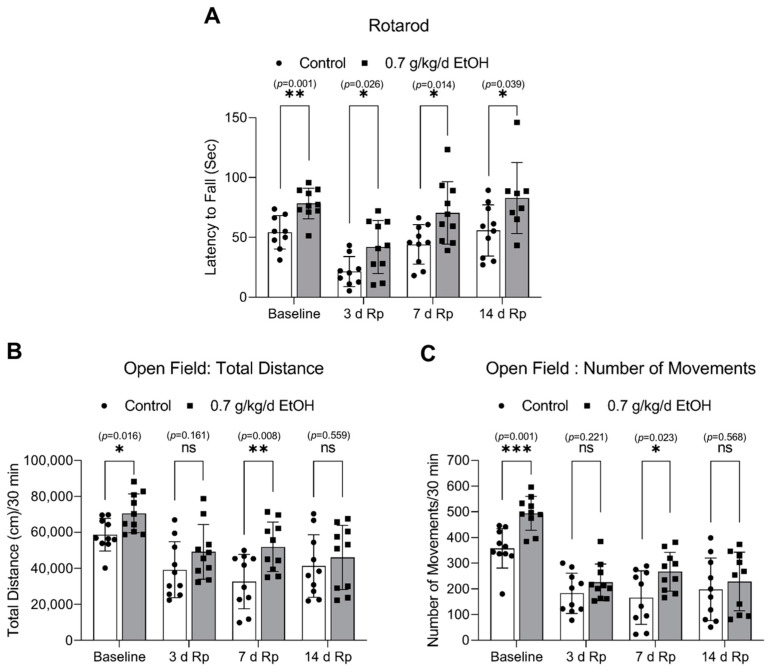
Effect of LAC on the locomotor activity under physiological conditions and following transient focal cerebral ischemia. (**A**) The latency to fall from the accelerating rotarod. (**B**) The total distance traveled during the 30-min trial in the open field test. (**C**) The number of movements during the 30-min trial in the open field test. Values are means ± SD (*n* = 10); * *p* < 0.05; ** *p* < 0.01; *** *p* < 0.005 vs. Control. Analyzed using an unpaired *t*-test.

**Table 1 biomedicines-11-01074-t001:** Effects of LAC on body weight, MABP, heart rate, and fasting blood glucose. Values are means ± SD for 6–20 mice in each group. Analyzed using an unpaired *t*-test.

	Control	0.7 g/kg/d EtOH	*p* Value
Body weight (g)	26.6 ± 2.0 (*n* = 20)	26.7 ± 1.4 (*n* = 20)	0.89
MABP (mmHg)	88.6 ± 13.3 (*n* = 10)	87.0 ± 14.2 (*n* = 10)	0.78
Heart rate (bpm)	653 ± 106 (*n* = 10)	598 ± 100 (*n* = 10)	0.23
Fasting blood glucose (mg/dL)	139.3 ± 19.7 (*n* = 6)	139.5 ± 26.5 (*n* = 6)	0.99

## Data Availability

The datasets generated and analyzed during the current study are available from the corresponding author on reasonable request.

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
