# Peer review of "Light Alcohol Consumption Promotes Early Neurogenesis Following Ischemic Stroke in Adult C57BL/6J Mice"

_biomedicines, 2023, doi:10.3390/biomedicines11041074_

Round 1
Reviewer 1 Report
In this manuscript, Li et al test the impact of light alcohol consumption on adult neurogenesis in multiple brain regions and following the MCAO model of stroke. They claim that the overall effect of LAC is to increase Dcx and NeuN co-expression with BrdU in both normal animals and following MCAO. Overall the premise is somewhat unusual and appears to be novel, however the execution is inadequate to allow the authors to reach the conclusions they draw.
The quantifications of the BrdU co-localizations do not appear to be accurate. The magnification and exposures of the representative images makes it impossible to tell if the cells that are claimed to be co-localized are in fact double positive. In fact, many of the examples presented by the authors as double-positive appear to not actually be (e.g. control in Fig 1C, all of Fig 2C, 4C). At minimum confocal imaging and higher magnification objectives are required to be certain of co-localization. The BrdU immunostaining also contains a lot of background, so it is not clear how much of the signal presented is real. The use of cresyl violet to assess infarct size is dubious and at least one alternative method should be considered. The behavior data while interesting is also not presented in ways that are typical for similar investigations and is similarly unconvincing, particularly without corroborating details and individual data points shown.
Author Response
Response to Reviewer 1 comments
- The quantifications of the BrdU co-localizations do not appear to be accurate. The magnification and exposures of the representative images makes it impossible to tell if the cells that are claimed to be co-localized are in fact double positive. In fact, many of the examples presented by the authors as double-positive appear to not actually be (e.g. control in Fig 1C, all of Fig 2C, 4C). At minimum confocal imaging and higher magnification objectives are required to be certain of co-localization. The BrdU immunostaining also contains a lot of background, so it is not clear how much of the signal presented is real.
We apologize for the unclear representative images. In the last version, we selected pictures with a solid background for showing the SVZ. In addition, NeuN staining in the SVZ and DG is always light because of the low maturity of newly generated neurons in these two areas. Therefore, it is hard to identify whether the cells are false double positives. We realized this problem during the experiment. Therefore, to exclude the possibility in these two areas, we only considered round-shape staining as BrdU staining and only the shape different from the BrdU staining as NeuN staining. Finally, the merged staining was considered as BrdU+/NeuN+ cells. In this revision, we have replaced the representative images with low background ones. In addition, we have provided 60x magnification images as supplemental data.
- The use of cresyl violet to assess infarct size is dubious and at least one alternative method should be considered.
The severity of the ischemic damage is one of the significant factors affecting neurogenesis. The purpose of measuring ischemic injury was to exclude the possibility that LAC-induced neurogenesis is related to exacerbated ischemic brain damage. TTC staining and Cresyl violet staining are extensively used to assess infarct size at the early stage of ischemic strokes, such as 24 hours and 48 hours of reperfusion. We previously used these two methods to determine the infarct size and consistently found that LAC significantly reduced the infarct size at 24 hours of reperfusion in the same MCAO model. However, neither technique showed infarction at two weeks of reperfusion. Furthermore, there is no area absenting DAPI staining under microscopy. Therefore, damaged tissue may be removed at two weeks of reperfusion to make the ischemic hemisphere smaller than the non-ischemic hemisphere. In the present study, we confirmed further that LAC reduced the I/R damage in the same brains, in which post-ischemic neurogenesis increased. Therefore, LAC-induced neurogenesis is not related to exacerbated ischemic brain damage.
- The behavior data while interesting is also not presented in ways that are typical for similar investigations and is similarly unconvincing, particularly without corroborating details and individual data points shown.
We have found that LAC promotes cerebral angiogenesis and neurogenesis. In addition, LAC typically protects motor function at the early reperfusion stage. Therefore, we decided to evaluate motor function more accurately using the accelerating rotarod and open field tests. We have replaced the line graphs with bar graphs, in which individual data points are shown. Locomotor activity was unexpectedly going down. We may need to study the effect of post-ischemic/in-hospital alcohol cessation in the future.
We would like to thank this Academic Editor for his/her helpful comments.
Reviewer 2 Report
This article discussed the impact of light alcohol consumption on neurogenesis in adult mice. There are some concerns in this manuscript that should be addressed as follows:
1. Title: It is preferable to mention the type of mice.
2. Key words: The word "mice" should be added to the key words.
3. The novel points in this study should be clarified because there are previous studies that discussed a similar topic.
4. The first two paragraphs of the Introduction have a very little number of references (i.e. 5 references only). Please, add more references.
5. The code of the ethical approval should be mentioned.
6. Page 2 line 79: Please add "ethyl alcohol" after " were fed with 0.7 g/kg/day "
7. A reference for the dose of ethyl alcohol should be mentioned.
8. The housing conditions of the animals used in this study should be mentioned.
9. The method used for determination of blood glucose levels should be mentioned.
10. How did the authors know that the animals were acclimatized?
11. The catalog numbers of the used kits and chemicals should be added.
12. I think that performing electron microscopic examination of the neurological tissue may add value to the present study.
13. Page 4: More details about the methods of statistical analysis should be added.
14. Figure 3: Arrows that indicate the positive histopathological findings should be added.
15. A collective diagram that summarizes the main findings of the present study should be added.
16. The conclusion should include the possible clinical applications of the results of the present study.
17. The manuscript should be checked regarding the grammatical and typing errors.
Author Response
Response to Reviewer 2 comments
- Title: It is preferable to mention the type of mice.
We have added the type of mice to the title.
- Key words: The word "mice" should be added to the key words.
We have added “mice” as a keyword.
- The novel points in this study should be clarified because there are previous studies that discussed a similar topic.
We have added “following ischemic stroke” to the title.
- The first two paragraphs of the Introduction have a very little number of references (i.e. 5 references only). Please, add more references.
We have added more references.
- The code of the ethical approval should be mentioned.
We have added the code and date of approval.
- Page 2 line 79: Please add "ethyl alcohol" after " were fed with 0.7 g/kg/day "
We have added it.
- A reference for the dose of ethyl alcohol should be mentioned.
We have added a reference to discuss the dose of ethyl alcohol.
- The housing conditions of the animals used in this study should be mentioned.
We have added the housing conditions of the animals used in this study.
- The method used for determination of blood glucose levels should be mentioned.
We have added a reference. In addition, we added a sentence to describe how the fasting blood glucose was measured.
- How did the authors know that the animals were acclimatized?
The trained investigators, the attending veterinarian, and his staff in Animal Resource monitored the animals.
- The catalog numbers of the used kits and chemicals should be added.
We have added the catalog numbers.
- I think that performing electron microscopic examination of the neurological tissue may add value to the present study.
Thank you for this suggestion. We may use an electron microscope to measure LAC-induced microstructural changes in the brain in the future.
- Page 4: More details about the methods of statistical analysis should be added.
We have modified the paragraph describing the methods of statistical analysis. We have added P values to the table and figures. In addition, we added the method of statistical analysis to the figure legend.
- Figure 3: Arrows that indicate the positive histopathological findings should be added.
We have added the arrows.
- A collective diagram that summarizes the main findings of the present study should be added.
We added a paragraph summarizing the main findings in the Results section. In addition, the main findings have been translated and discussed in the Discussion section.
- The conclusion should include the possible clinical applications of the results of the present study.
We have added the possible clinical applications of the present study to the summary paragraph.
- The manuscript should be checked regarding the grammatical and typing errors.
We have checked the grammatical and typing errors.
We would like to thank this reviewer for his/her helpful comments.
Round 2
Reviewer 1 Report
In the revised version of the manuscript the authors have provided new confocal images of their immunostaining data. These new data more clearly demonstrate several the issues with the original study: 1) The authors still present co-stained cells that are not co-stained (e.g. arrow on the left in Supplementary Fig 2 EtOH). This continues to call into question the accuracy of all of the quantifications presented. The count data are presented as fold-change which could mean they counted as few as 5 or 10 cells per-animal based on representative images with 1 or 2 cells and the statement that they assessed 5 sections in total. 2) the NeuN staining is weak and in many instances questionably above background in the confocal images which the authors claim could be due to the immature nature of the cells produced, but they provide no evidence to support this assumption. 3) Cells they present as co-stained are deep in the hilus which could be interesting if confirmed, but without additional evidence is unsupported by the literature. 4) Only one time point is examined making broader claims about neurogenesis difficult to support.
Beyond all of these methodological issues is a broader one of the over-reaching interpretations. The individual data points make it very clear that LAC did not “improve locomotor activity considerably”. Only a few of the animals tested exhibited differences from the control groups, and only in some comparisons. Pair-wise t-tests are not the appropriate statistical test.
Overall this paper reports one experiment with one time-point that includes undersampling and statistical inaccuracies suggesting that it is unlikely to be reproducible.
Author Response
1) The authors still present co-stained cells that are not co-stained (e.g. arrow on the left in Supplementary Fig 2 EtOH). This continues to call into question the accuracy of all of the quantifications presented. The count data are presented as fold-change which could mean they counted as few as 5 or 10 cells per-animal based on representative images with 1 or 2 cells and the statement that they assessed 5 sections in total.
We apologize for the mistake made on the supplementary Figure 2 during the revision. We have removed that arrow. Also, we have carefully checked all indicators on other representative images. We picked up the representative photos after finishing the quantitative analysis. Therefore, the mistake does not affect the quality of the present study. Removing the arrow improved the representative images since the supplementary Figure 2 represents a non-statistically significant change between control and ethanol. This reviewer mainly has a concern about BrdU+/NeuN+ co-stained cells. We will clarify his/her concern in the following response. Indeed, BrdU+/DCX+ and BrdU+/NeuN+ cells were only counted as few as 5 or 10 cells in the particular section under physiological conditions. The present study did not count BrdU+/DCX+ or BrdU+/NeuN+ cells in the entire SVZ, DG, ischemic cortex, or ischemic striatum. Thus, fold-change may be a better way to express this type of measurement since there is no appropriate unit for the Y axle.
2) the NeuN staining is weak and in many instances questionably above background in the confocal images which the authors claim could be due to the immature nature of the cells produced, but they provide no evidence to support this assumption.
We have added references and discussion on this concern. There are strongly NeuN-stained BrdU-positive neurons (mature neurons) in the DG and SVZ, suggesting that neurons do mature in these two areas. In addition, a few previous studies found that developing neurons express NeuN lightly. Since there is no standard to determine light and heavy staining, we included both heavily and lightly NeuN-stained BrdU-positive (BrdU+/NeuN+) cells as newborn neurons to indicate neurogenesis. However, we were concerned about identifying whether the light staining of NeuN is a background or represents the maturing process. Therefore, we only considered round-shape staining as BrdU staining, and the shape differed from the BrdU staining as NeuN staining. In addition, the increase of BrdU+/DCX+ cells also indicates that LAC promotes neurogenesis.
3) Cells they present as co-stained are deep in the hilus which could be interesting if confirmed, but without additional evidence is unsupported by the literature.
In the present study, we aimed to find whether LAC affects co-stained newborn neurons in the DG. Thus, it is beyond the scope of the present study whether the co-stained cells are deep in the hilus.
4) Only one time point is examined making broader claims about neurogenesis difficult to support.
We have added “early” to the title. In addition, we have added two references. Neurogenesis peaks two weeks after the onset of ischemic stroke. Thus, we selected two weeks to measure post-ischemic neurogenesis in the present study.
5) Beyond all of these methodological issues is a broader one of the over-reaching interpretations. The individual data points make it very clear that LAC did not “improve locomotor activity considerably”. Only a few of the animals tested exhibited differences from the control groups, and only in some comparisons. Pair-wise t-tests are not the appropriate statistical test.
We used unpaired t-tests for all comparisons in the present study. We agree with this reviewer that LAC did not improve locomotor activity considerably. Therefore, we have removed “considerably” from the abstract.
We would like to thank this reviewer for his/her helpful comments.
Reviewer 2 Report
The authors had appropriately addressed most of my comments
Author Response
We have checked spelling and grammar.
We would like to thank this reviewer again for his/her helpful comments.